# Cardiac Autonomic Function in Long COVID-19 Using Heart Rate Variability: An Observational Cross-Sectional Study

**DOI:** 10.3390/jcm12010100

**Published:** 2022-12-22

**Authors:** Antonio da Silva Menezes Junior, Aline Andressa Schröder, Silvia Marçal Botelho, Aline Lazara Resende

**Affiliations:** 1Internal Medicine Department, Medicine School, Federal University of Goiás, Goiânia 74175-120, Brazil; 2Medical and Life Sciences School, Medicine School, Pontifical Catholic University of Goiás, Goiânia 74000-000, Brazil

**Keywords:** heart rate variability, inflammatory markers, long-term COVID-19, autonomic nervous system

## Abstract

Background: Heart rate variability is a non-invasive, measurable, and established autonomic nervous system test. Long-term COVID-19 sequelae are unclear; however, acute symptoms have been studied. Objectives: To determine autonomic cardiac differences between long COVID-19 patients and healthy controls and evaluate associations among symptoms, comorbidities, and laboratory findings. Methods: This single-center study included long COVID-19 patients and healthy controls. The heart rate variability (HRV), a quantitative marker of autonomic activity, was monitored for 24 h using an ambulatory electrocardiogram system. HRV indices were compared between case and control groups. Symptom frequency and inflammatory markers were evaluated. A significant statistical level of 5% (*p*-value 0.05) was adopted. Results: A total of 47 long COVID-19 patients were compared to 42 healthy controls. Patients averaged 43.8 (SD14.8) years old, and 60.3% were female. In total, 52.5% of patients had moderate illness. Post-exercise dyspnea was most common (71.6%), and 53.2% lacked comorbidities. CNP, D-dimer, and CRP levels were elevated (*p*-values of 0.0098, 0.0023, and 0.0015, respectively). The control group had greater SDNN24 and SDANNI (OR = 0.98 (0.97 to 0.99; *p* = 0.01)). Increased low-frequency (LF) indices in COVID-19 patients (OR = 1.002 (1.0001 to 1.004; *p* = 0.030)) and high-frequency (HF) indices in the control group (OR = 0.987 (0.98 to 0.995; *p* = 0.001)) were also associated. Conclusions: Patients with long COVID-19 had lower HF values than healthy individuals. These variations are associated with increased parasympathetic activity, which may be related to long COVID-19 symptoms and inflammatory laboratory findings.

## 1. Introduction

The severe acute respiratory syndrome-related coronavirus-2 (SARS-CoV-2) virus is the source of the pandemic known as the coronavirus disease 2019 (COVID-19). This pandemic has been responsible for a significant amount of illness, as well as mortality and disruption. Now, the primary focuses of healthcare practitioners are on the prevention of new instances and the development of rehabilitation regimens [1].

According to World Health Organization (WHO), the post-COVID-19 condition develops in people with a history of suspected or proven SARS-CoV-2CoV-2 infection three months after COVID-19 infection with symptoms that continue for at least two months and cannot be explained by another diagnosis. Fatigue, shortness of breath, cognitive impairment, and other symptoms affect daily life. After recovering from COVID-19, symptoms may be new or remain. Symptoms may change or recur [2]. This disease may cause pulmonary, cardiac, and vascular fibrosis, plus fatigue [1,3]. Scientific, medical, and social institutions recognize this disorder, which affects COVID-19 survivors across various ages and illnesses, including young people and those not hospitalized [1,2]. However, this condition is still poorly understood [2,3,4,5,6,7,8,9,10].

Dysautonomia is characterized by a change in the activity of the autonomic nervous system (ANS), manifesting in various ways including fatigue, labile blood pressure, orthostatic hypotension, and heart rate variability (HRV) dysfunction. It is a non-invasive metric of cardiovascular autonomic regulation that is referred to as HRV, which stands for the variability that occurs between two consecutive heart beats. The traditional method for calculating HRV involves first getting heart rate (HR) data from an electrocardiogram (ECG), and then doing the computation using specialist computer software [5]. It is related to several conditions, including metabolic syndrome, diabetes, and neurological disorders, and may manifest as acute or chronic dystonia with progressive, reversible symptoms [3,4,5,6]. Viral illnesses are a source of dysautonomia [7]. In long COVID-19 patients, dysautonomia relates to significant fatigue, orthostatic intolerance, and orthostatic hypotension [7,8,9,10].

COVID-19 and long COVID-19 may be systemic disorders linked with inflammation and a procoagulant condition, and they can induce parasympathetic and sympathetic imbalances with acute and convalescent signs and symptoms. Despite the significant number of individuals who suffer from the post-COVID-19 syndrome, only a small amount of research has been conducted on this topic thus far [11,12,13,14].

We aimed to noninvasively measure HRV dysregulation in long COVID-19 patients, reflected by low-frequency (LF) and high-frequency (HF) indices determining sympathetic/parasympathetic equilibrium failure. We hypothesize that the symptoms of late COVID-19 are directly connected to a virus or ANS dysregulation associated with the immune system, resulting in acute or long-term dysautonomia. Dysautonomia assessment may help to monitor the severity of long COVID-19 syndrome.

## 2. Patients and Methods

### 2.1. Study Design and Setting

The study was an observational, cross-sectional study on long COVID-19 patients examined at the Medical Education and Clinical Research Center in Brazil between August 2020 and June 2021.

### 2.2. Participants

#### 2.2.1. Inclusion Criteria

Patients with long COVID-19 syndrome after 3 months of mild, moderate, and severe COVID-19 (mild cases of COVID-19 were defined as those who were symptomatic but did not have evidence of major viral pneumonia or hypoxia; moderate cases were defined as those who had clinical signs of pneumonia but did not have signs of complications, such as SPO2 90% in room air, and severe cases were defined as those who had clinical signs of pneumonia and signs of complications, such as acute respiratory distress syndrome, sepsis, thromboembolic events, and pulmonary involvement which were included in the study. In order to prevent possible sources of bias in the analysis of HRV, healthy patients with a negative reverse transcription–polymerase chain reaction (RT-PCR) test result and no symptoms were included in the study as a control group.

The COVID-19 status of all inpatients was determined via RT-PCR testing employing a nasopharyngeal and oropharyngeal swab and a polymerase chain reaction (PCR) kit. An appropriate selection approach was utilized to select eligible male and female inpatients (test group) aged 18–70 years for the sample.

#### 2.2.2. Exclusion Criteria

Patients who were using beta-blockers, beta-mimetics (inhaled or oral), theophylline, or any other medication that had potential chronotropic effects were excluded from the analysis.

#### 2.2.3. Procedures

In our medical outpatient clinic, we monitor post-COVID-19 patients as well as clinical patients with or without comorbidities. Patients were chosen based on whether they met the inclusion criteria or not. This created two groups of patients: those with the post-COVID-19 syndrome and those without COVID-19 and, therefore, the post-COVID-19 syndrome.

For cases, we performed the following: comprehensive clinical evaluation (symptoms questionnaire); a 24 h ambulatory electrocardiogram; computed tomography (CT) of the chest; biochemical tests for markers of inflammatory processes, such as ultrasensitive C-reactive protein (CRP) dosage, procalcitonin, calcitonin, D-dimer, interleukin 6, and serum ferritin. For control patients, a 24 h ambulatory electrocardiogram was performed.

### 2.3. Heart Rate Variability Assessment

Both groups were instructed to refrain from smoking, caffeine intake, and alcohol intake for 36 h prior to the investigation. HRV, a quantitative index of autonomic activity was recorded for 24 h using an ambulatory ECG system (DMS 300-4L Holter ECG Recorder Los Angeles, CA, USA). Prominent features include 3 channels of continuous ECG, multiple electrode size options, tap–tap event markers, and an internal clock.

The stabilization period prior to sampling was observed in all analyses of sample collection. The acknowledgment of breathing was controlled. Holter (24 h ECG evidence) was previously analyzed and was checked by two independent physicians, not knowledgeable of the group the patient belonged to, the artifacts and percentage of beats were corrected. The frequency band specification was calculated from the Fast Fourier Transform.

In the temporal domain, measures of SDNN, rMSSD, and the proportion of successive RR periods exceeding 50 ms (pNN50) were studied. While measuring power in the frequency domain, it was possible to estimate the distribution of absolute or relative power into separate frequency bands, as well as LF and HF domain measurements.

The parasympathetic component was represented by the HF component, whereas the sympathetic component was represented by the LF component. Furthermore, the parasympathetic activity was quantified by SDNN, rMSSD, and pNN50 components [11,15,16,17,18], as seen in Table 1.

### 2.4. Statistical Analysis

The GPower^®^ 3.1 software (Heinrich-Heine-Universität, Düsseldorf, German) was used to estimate sample size. The following parameters were used: the types of tests selected were the chi-square test (contingency tables), the Student’s t test (comparison of means for two independent samples), and the F test (one-way ANOVA). The following parameters were also defined: effect size (W = 0.3), alpha probability error (=0.05), and power (error probability 1 = 0.95). The dataset was submitted for descriptive and inferential statistics. For categorical variables, absolute and relative percentage frequencies were calculated, as well as mean (central trend measure) and standard deviation (dispersion measure) for continuous variables. In the inferential statistics, the chi-square and G tests were applied to compare the case and control groups for the categorical variables. Additionally, a D’Agostino–Pearson normality test was applied to verify the distribution of the data. The means were compared with the Student’s *t*-tests for two independent samples (for dichotomous variables) and one-way ANOVA (for polytomic variables). BioEstat 5.3 software (AnalystSoft Inc., Walnut, CA, USA) was used for this. Multivariate logistic regression was used to analyze comorbidities vs. COVID-19 and the effect was estimated with the calculation of an odds ratio (OR) with a 95% confidence interval, using Stata^®^ 16.0. (StataCorp. 2019. Stata Statistical Software: Release 16. College Station, TX, USA: StataCorp LLC). A significance level of 5% (*p*-value 0.05) was adopted.

### 2.5. Ethical Approval

This study was conducted in accordance with the principles embodied in the Declaration of Helsinki. Following CNS Resolution 466/2012, the Research Ethics Committee approved the study (approval no. CAAE 47544021.9.0000.0037). All study participants provided written informed consent, which was verified by the research ethics committee.

## 3. Results

The study included 47 COVID-19-positive (previous RT-PCR positive) patients who completed a questionnaire about their symptoms and relationships and supplemental tests (test group). In total, 42 people who tested negative for COVID-19 (RT-PCR) (control group) had a 24 h ambulatory ECG (the entire 24 h sample was used for all the metrics) (Table 2), as seen in the central figure.

The average age of patients was 43.8 years (SD ± 14.8 years), and females accounted for 60.3%. Patients aged >40 years formed the majority of those interviewed (53.2%). A modest COVID-19 profile was seen in most patients (52.5%), and the majority (97.9%) received diagnostic confirmation via a detectable RT-PCR test. An average of 3.6 months (2.6 months) passed between infection and the primary consultation (interview using a questionnaire to describe the symptoms and signs). The treatment was largely performed at home (73%), and >25% of pulmonary involvement was detected in 34% of patients, with a mean degree of impairment of 34.4% (29%) in the most severe cases.

The case group had few comorbidities (53.2%). There was no statistically significant difference between case patients and the controls (Table 1).

Mild, moderate, and severe profiles of COVID-19 had considerably greater (*p* 0.0001) pulmonary involvement on chest tomography, with a mean impairment of 4.3%, 14.2%, and 25.8%, correspondingly. BNP, D-dimer, and CRP also showed statistical significance, with *p*-values of 0.0098, 0.0023, and 0.0015 during the long COVID-19 evaluation, respectively (Table 3).

SDNN-24 and SDANNi correlated positively (r = 0.9852). rMSSD and pNN50 were strongly correlated (r = 0.9986). Case and control groups had different mean and lowest heart rates. Another significant difference was heart rate (*p* = 0.0297). Long COVID-19 periods were associated with higher mean heart rates (OR = 1.07 (1.02 to 1.13), *p* = 0.04). Higher numbers of SDNN-24 and SDANNi were found in the control group (OR = 0.98 (0.97 to 0.99; *p* = 0.01) for each). Table 4 shows a link between elevated LF in COVID-19 patients (OR = 1.002 (1.0001 to 1.004), *p* = 0.03) and HF in the control group (OR = 0.987 (0.98 to 0.995), *p* = 0.001) (Table 4).

## 4. Discussion

Cardiorespiratory symptoms were the most common long COVID-19 symptoms in our investigation, indicating that these organs may be involved even beyond the acute phase of the illness [4,19]. CT showed that lung involvement was statistically significant compared to symptom severity, suggesting that CT is effective in evaluating pulmonary injury. BNP, D-dimer, and CRP detect inflammation in various disorders [14]. These inflammatory indicators are crucial for evaluating and monitoring patients with acute and long-term COVID-19, notably CRP, which had lower *p*-values. Multiple lines of evidence show the prognostic utility of this biomarker, which has been the focus of many COVID-19 case publications. Clinical investigations of CRP demonstrate a positive connection between sickness severity and baseline CRP levels [20].

Knowing that CRP is a prognostic marker of inflammation, one research correlated a rise in CRP levels and the disease’s inflammatory levels with HRV in these individuals. In long-COVID-19 patients, a 40% drop in HRV followed a 50% rise in CRP [21]. In our research, the case group had lower HRV, contributing to the idea that they had more inflammation.

COVID-19 may be responsible for the perpetuation of arrhythmias due to an increased catecholaminergic state caused by cytokines such as IL-6, IL-1, and tumor necrosis factor-alpha. This increased catecholaminergic state may prolong ventricular action potentials by altering the expression of cardiomyocyte ion channels [22,23,24,25]. Atrial fibrillations might be caused by COVID-19 (AF). During a viral infection, adrenergic modulation contributes to the development of postural orthostatic tachycardia syndrome as well as an irregular sinus rhythm. Our study found a connection between features of heart rate and dysautonomia [26,27,28,29,30].

Heart rate variability (HRV) is a marker of cardiac dysautonomia. In order to provide evidence in favor of the dysautonomia hypothesis, we investigated heart rate variability (HRV) in both the time and frequency domains by performing a non-invasive study of sympathetic and parasympathetic activity in long COVID-19 patients. It is possible that HRV may be a useful tool for examining neuroimmune systems and inflammatory processes of long COVID-19 patients. The high-frequency (HF) component of spectrum analysis and the LF/HF ratio, both of which are methods to assess vagal dysfunction, have been related to the long COVID-19 syndrome. In individuals with long COVID-19, dysautonomia may have a variety of causes, including neurotropism, procoagulation, and inflammation. It is not known whether the protracted COVID-19 dysautonomia is caused by immune-mediated mechanisms after infection or by the autonomic–virus pathway. For the SARS-CoV-2 virus to enter cells during the acute phase of infection, angiotensin-converting enzyme 2 (ACE2) is required.

In our study, SSDNN-24, SDANNi, rMSSD, pNN50, Max QTc, and Max QT showed substantial differences. When comparing data in the time-domain and HRV frequency-domain, a high positive association (r = 0.9986) was seen between SDNN-24 and SDANNi. Employing HRV measures in the time-domains rMSSD and SDNN better indicated the increase in parasympathetic activity in individuals with long COVID-19 and autonomic dysfunction [9]. Another study found that HRV, as measured by the NOL index, was substantially different between patients with long COVID-19 and tiredness and the control group (without a diagnosis of COVID-19 and fatigue) [4].

Aragn-Benedi et al. [26] found a prevalence of a parasympathetic tone and a concomitant withdrawal of sympathetic activity, along with a decline in HRV in COVID-19-complicated patients, with the latter finding more prominent in those with a poorer outcome. According to the author, this finding is due to the pathogenic, cholinergic, and anti-inflammatory response that results from sympathetic overactivity, causing immunological anergy and a poorer prognosis. Pan et al. [27] observed an increase in sympathetic activity during the initial phase of infection. They found a drop in HRV in critically sick patients in terms of SDNN and SDANN, as well as an increase in LF/HF, which coincided with elevations in humoral biomarkers such as NT-proBNP and D-dimer.

Hasty et al. [21] found a drop in HRV characteristics in moderate COVID-19 individuals, which predicted a 72 h increase in CRP. In severe phases of the disease, sympathetic activity may explain the disparities between the two studies. After a systemic inflammatory response syndrome is diagnosed, the sympathetic–vagal balance may shift towards parasympathetic dominance to minimize systemic inflammation.

Infection with COVID-19 causes a “cytokine storm” with increased CRP and IL-6 [25], and these predict disease severity and poorer outcomes. The ANS regulates inflammation. Increased vagal responses boost HRV and decrease inflammation through the cholinergic anti-inflammatory pathway [26]. Pro-inflammatory sympathetic hyperactivity decreases HRV and has detrimental effects. Low HRV and dysautonomia have been linked to HIV, CAP, and Dengue [27,28,29]. Short- and long-term HRV recordings are inversely related to inflammatory markers [30,31]. In this study, a protracted COVID-19 duration was linked to a higher mean heart rate. Higher numbers of SDNN 24 and SDANNI in control patients (OR = 0.98 (0.97 to 0.99; *p* = 0.01)) were associated.

Higher LF in long COVID-19 patients (OR = 1.002 (1.0001 to 1.004), *p* = 0.03) was associated with increased HF in the control group (OR = 0.987 (0.98 to 0.995), *p* = 0.001). Dysautonomia may explain long COVID-19 patients’ symptoms, according to an HRV power spectrum study. Vagal dysfunction is caused by procoagulation and cardiac strain. Long COVID-19 fiber loss may suggest vagal dysfunction [30]. Long COVID-19 autonomic innervation in vivo suggests clinical relevance. Appropriate medication may improve long-term prognoses for long COVID-19 patients, who must be examined for diminished vagal activity, protracted NT-ProBNP elevations, and prothrombotic circumstances [31,32,33,34,35].

## 5. Limitations

Our study’s limitations include the following: small sample size, all the limitations and risk of bias inherent to cross-sectional study designs, the inability to generalize the findings to different populations, and the potential for selection bias due to the single-center design.

## 6. Conclusions

Long COVID-19 is associated with a preponderance of parasympathetic activity in autonomic heart rhythm control. This is a study to assess HRV in long COVID-19 patients utilizing frequency and spectrum analysis. Dysautonomia may explain long COVID-19 patients’ symptoms. Vagal dysfunction in these people may be caused by procoagulation and heart strain.

The impairment of cholinergic nerve fibers in individuals with long COVID-19 may indicate that the vagus nerve is dysfunctional. In our laboratory, we are determining whether or not individuals with long COVID-19 have abnormal autonomic innervation. Our research may have applications in the medical field. Treatment that is appropriate has the potential to improve both the clinical presentation and prognosis of long COVID-19 patients. Patients on the long COVID-19 protocol need to be carefully watched for signs of decreased vagal activity, prolonged elevations in NT-ProBNP, and a prothrombotic condition.

## Figures and Tables

**Table 1 jcm-12-00100-t001:** Heart rate variability and their equivalent representation of the autonomic nervous system.

Acquisition	System of Measurement	Category	Autonomic Reflection
Time-domain	SDNN	The standard deviation of all normal–normal (R–R) intervals	PNS and SNS activity
	pNN50	Percentage of consecutive N–N intervals that deviate from one another by more than 50 ms	PNS activity
	RMSSD	The square root of the mean squared differences between normal adjacent R–R intervals	PNS activity
Frequency-domain	TP	Total power (<0.4 Hz)	Variability in autonomic function as a complete
	VLF	Very low frequency (<0.04 Hz)	Thermoregulatory cycles
	LF	Low frequency (0.05–0.15 Hz)	Combined action of the PNS and SNS
	HF	High frequency (0.15–0.4 Hz)	PNS activity
	LF: HF	The ratio of low-frequency to high frequency	SNS-to-PNS balance

PNS: the parasympathetic nervous system, and SNS: the sympathetic nervous system. Source: [5].

**Table 2 jcm-12-00100-t002:** Demographic characteristics of the case and control groups.

Variable	Case Group (*n* = 47)		Control Group (*n* = 42)		*p*-Value
	*n*	%	*n*	%	
Comorbidities					
Arterial hypertension	8	17.0	10	23.8	
Dyslipidemia	7	14.9	3	7.1	
Obesity	4	8.5	5	11.9	
Diabetes mellitus	2	4.3	2	4.8	
Chagas	1	2.1	1	2.4	
None	25	53.2	21	50.0	0.8426
Sex					
Female	28	59.6	30	71.4	
Male	19	40.4	12	28.6	0.2413

**Table 3 jcm-12-00100-t003:** Descriptive statistics of the case group by disease severity (COVID-19).

Variables (*n* = 47)	Mild (*n* = 20)		Moderate (*n* = 19)		Severe (*n* = 6)		*p*-Value
	Mean	SD	Mean	SD	Mean	SD	
Age (years)	41.2	10.3	46.3	11.7	47.5	17.0	0.3132
Time * (months)	4.2	2.3	5.2	2.2	4.7	1.9	0.5787
Chest CT ** (%)	4.3	6.3	14.2	9.3	25.8	10.2	<0.0001
Echocardiography LVEF (%)	63.2	5.0	58.7	6.4	58.0	8.9	0.0481
BNP (pg/mL)	15.2	12.4	35.0	43.2	44.8	20.9	0.0098
Calcitonin (pg/mL)	2.7	1.0	3.2	1.8	3.9	2.3	0.3642
D-dimer (ng/mL)	180.8	121.2	312.9	221.0	454.4	179.5	0.0023
Ferritin (pmol/L)	209.4	164.6	302.4	232.5	365.8	274.9	0.2102
CRP (mg/L)	3.4	2.8	3.7	2.9	8.9	4.5	0.0015
Procalcitonin (ng/mL)	0.4	0.7	1.3	1.9	0.4	0.3	0.1976
Fibrinogen (mg/dL)	358.5	163.5	368.4	179.3	454.2	183.6	0.5100
IL-6 (pg/mL)	3.4	1.4	4.1	1.5	4.0	1.8	0.5921

CT: computed tomography; LVEF: Left Ventricular Ejection Fraction; BNP, brain natriuretic peptide; CRP, C-reactive protein; IL-6: interleukin-6.; * The time between the patient’s serological test and hospital discharge, and the selection based on the inclusion criteria to participate in the study ** Areas of ground-glass opacity; Standard values: BNP: 0 to 70 pg/mL; Calcitonin < 10 pg/mL; D-Dimer: <350 ng/mL; Ferritin: 30 to 300 ng/mL (67.4 to 674.1 pmol/L); Procalcitonin: >0.500 ng/mL; CRP: 0.3 mg/dL (ou 3 mg/L); Fibrinogen: 200–400 mg/dL; IL-6: 1.5 to 7.0 pg/mL.

**Table 4 jcm-12-00100-t004:** Comparisons between the case and control groups.

Variable	Case (*n* = 47)		Control (*n* = 44)		*p*-Value
	Mean	SD	Mean	SD	
Age	44.4	12.2	39.6	12.9	0.0709
HR	82.3	9.2	75.8	10.0	0.0018
Min HR	52.4	11.5	48.1	9.3	0.0253
Max HR	130.9	18.9	125.6	19.2	0.1862
VE	267.7	1533.6	126.6	494.3	0.7060
SVE	90.6	419.1	12.3	36.5	0.9335
SDNN-24	111.6	38.7	133.4	37.8	0.0078
SDANNi	99.7	38.5	122.3	39.9	0.0072
rMSSD	41.8	86.3	34.3	12.2	0.0310
pNN50	18.3	66.7	11.8	8.6	0.0442
Max QTc	544.6	101.8	518.9	44.0	0.0389
Max QT	503.9	77.2	467.5	40.5	0.0086
VLF	2225.7	1631.9	2342.1	1183.8	0.2398
LF	780.4	513.1	828.5	417.7	0.6262
HF	233.6	172.2	307.3	196.1	0.0297

SVE: supraventricular extrasystoles; VE: ventricular extrasystoles; SDNN: standard deviation of a normal sinus beat interbeat intervals (IBI); SDANNi: standard deviation of the average normal-to-normal (NN) intervals for each 5 min segment; rMSSD: root mean square of consecutive normal heartbeat disparities, determined by first calculating each subsequent time difference in ms; pNN50: proportion of neighboring NN intervals that vary by 50 ms, also required 2 min. The VLF band (0.0033–0.04 Hz) needs a recording time of at least 5 min, but may be best monitored over 24 h; the LF band (0.04–0.15 Hz) is usually recorded for 2 min.

## Data Availability

The datasets generated and/or analyzed during the current study are available from the corresponding author upon reasonable request.

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
