# Peer review of "Cardiac Autonomic Function in Long COVID-19 Using Heart Rate Variability: An Observational Cross-Sectional Study"

_jcm, 2022, doi:10.3390/jcm12010100_

Round 1

Reviewer 1 Report

This is an interesting study and does contribute to the long-COVID knowledge gap even though others have published on this topic already (HRV and long COVID).  However, there are many grammatical errors and serious mistakes that need to be corrected.  The use of logistical regression comparing patients with long-covid to controls that never had Covid is a serious flaw and needs to be addressed.  

Introduction - There are many definitions, but WHO definition is probably most widely used.  see->  https://www.who.int/publications/i/item/WHO-2019-nCoV-Post_COVID-19_condition-Clinical_case_definition-2021.1  This should be included in the introduction.  

"In COVID-19 patients, dysautonomia relates to significant fatigue [7–10]"  Is this the only symptom related to dysautonomia in long-COVID patients??  Orthostatic inolerance and orthostatic hypotension, conditions that were discussed in your paragraph above are also manifestations.  This sentence needs to be reworded to include all these symptoms.  

I would reocmmend removing the middle 3 paragraphs in the introduction section.  You don't need to explain why there is autonomic dysfunction in long COVID patients in this section; it's everywhere in the literature, simply cite a few sources.  Moreover, it is filled witih errors and inconsistencies.  Citation 11 is an article on heart rave variability values and does not address inflammatory or anti-inflammatory effects.  Is there an immune system/inflammation explanation to long COVID?  Possibly.  But it certinaly should not be discussed in the intro section.  

Section 2.3

Please include entire definition of pNN50.  It is the proportion of successive RR periods exceeding 50ms.

Last sentence:  "parasympathetic activity was quantified by" not "into"

Results

This sentence makes no sense.  "SDNN-24, SDANNi, rMSSD, pNN50, Max QTc, and Max QT showed statistically significant changes from SDANN"

The duration of monitoring for each HRV metric needs to be stipulated.  Were the entire 24h sample used for all the metrics?  Some only require 5 minutes of recorded time, some even less.  

Table 3

"Due to the reduced heart rate, the long COVID-19 group had a significant incidence of dysregulated parasympathetic activity (Table 3)"  what are you coming this reduced heart rate to? In table 3, the controls had a lower heart rate than you COVID patients.  

Logistical regression and Table 4

"Long COVID-19 was strongly associated with dyslipidemia (OR = 3.9 (1.2 to 12.57; p = 0.02)), which was 4 times more common in COVID-19 carriers than in controls (Table 4)"  

There is not a lot of utility comparing comorbidities and other charactistics between patients who developed long-COVID to normal controls (who never had COVID).  The biggest risk factor to developing long COVID is having COVID in the first place, which none of your controls had.  A better comparison would be comparing pts who developed long-COVID to pts who had COVID but did NOT develope long-COVID.  I would remove table 4 and all its associated texts.  

Page 7, define ACE2

Discussion:

"HRV [31–35] is a legitimate dysautonomia diagnosis."  This sentence makes no sense.  HRV is a not a disease or condition, but a metric that can be used to evaluate the autonomic system.

"The high-frequency (HF) component of spectrum analysis and the LF/HF ratio, both of which are indications of vagal dysfunction"  This is incorrect.  They are metrics that assess parasympathetic tone.  They do not indicate vagal dysfunction; even patients with normal parasymptathetic tone will have signal in the HF spectrum.  

"The fact that there was less heart failure in the case group compared to the control group is suggestive of an increase in parasympathetic activity in the extended COVID group [."  This sentence makes no sense, please delete it.  

Conclusions:

"In our laboratory, we are determining whether or not individuals with prolonged COVID-19 have autonomic innervation."  Do you mean abnormal autonomic innvervation?  Everyone has autonomic innervation.   

Author Response

Dear Reviewer

 We would like to thank you for your thorough analysis of our article; we attempted to address each of the points raised below in an attempt to improve our potential publishing. All suggestions were critical to ensuring work with more scientific proof. We answered the pointed-out questions with great pleasure, which allows us to advance in the development of knowledge.

Reviewer 2 Report

Please note attachment. 

Author Response

This point was missing in Reviewer Response:

How high was the sample size determined with GPPower?

The sample size of the survey is relatively small. In attempt to get an estimate of the sample size, the program GPower® 3.1 was used. These are the parameters that were utilized: the chi-square test (contingency tables), the Student's t test (comparison of means for two independent samples), and the F test (one way ANOVA). In addition, the following terms were defined: power (error probability 1-=0.95), alpha probability error (W=0.05), and effect size (W=0.3).

Reviewer 3 Report

The manuscript is very good and I am sure it will be of great interest to the scientific community involved in this field. However, I found some issues that should be corrected before being published.

In attachment, you will find the pdf where I make the observations.

Author Response

.

Reviewer 4 Report

In this study the authors explore a novel aspect of the prevalent and interesting long-COVID syndrome. They hypothesize that the symptoms can be partially explained by autonomous dysregulation mediated by the COVID infection. In order to test this hypothesis, they examine the heart rate variability in post COVID patients and received interesting hypothesis generating results.

 After reading this paper I have several unanswered questions. 

1. The title of the paper and the discussion all deal with long-COVID. The study design does not address this aspect. I could not understand how and when the patients were selected. Were they all hospitalized? When were they diagnosed? Were they symptomatic? How were they classified as long-COVID? when were blood tests taken? at what stage was the 24hr ECG done? Similarly, how were the control patients selected? was there any matching between controls and study patients?

2. No information is given about the control patient selection, therefore all comparisons between the baseline characteristics of the controls and the study group are irrelevant.

3. How were comorbidities collected? medical record? questionnaire? how was each comorbidity defined? were IHD and HF an exclusion criterion? how many were smokers (not listed in table 1)?

Regarding statistical methods and analysis:

1. Data presentation: Table 1 should include all demographical data - age, sex, % of mod and severe covid, time from diagnosis, symptoms at presentation etc. 

2. Table 2 is hard to understand. First, all rows should have units. Additionally, what do the rows chest CT and echo represent? tests done or abnormal findings? What are the normal limits for all the test results in your lab?

3. Results would have been much more meaningful if blood test were compared with healthy controls. 

4. Table 3 would be much more informative if the authors would add the results of the mild, mod and severe COVID cases.

5. How was sample size assessed? The authors gave no information about their preliminary assumptions. 

6. In table 4, are the OR value derived from multivariable analysis? if so, what were the values used for the analysis? 

7. In the introduction and discussion, most of the acronyms are not given upon first usage.  since there are many of them, consider listing them all at the beginning of the article. 

8. The introduction is long. The paragraph starting with "Owing to inadequate immune response control..." does not add to the paper and can be deleted. instead, the authors should briefly describe the reasoning behind HRV, and how can it be used to estimate sympathetic and para-sympathetic activity.

9. Was there any attempt to analyze the associations between blood markers and HR variables in this population?

Regarding study relevance:

1. The paper discusses long-COVID symptoms. however, the study patients were not defined as long-COVID patients. Therefore, the study aims should be modified. 

I think the paper represents a great idea with very interesting and relevant findings. However, I think there is a need for re-writing in order to clarify some of the findings

Author Response

We greatly appreciate your collaboration and attention to our article, all the points taken were pertinent, and in the case of possible, we try to accept the suggestions presented. These notes will make our research a greater degree of relevance and better scientific relevance.

Round 2

Reviewer 1 Report

In the introduction, this sentence

"In long COVID-19 patients, dysautonomia relates to significant fatigue, as also orthostatic intolerance, and orthostatic hypotension [7–10]."

Needs to be editted to this:

"In long COVID-19 patients, dysautonomia relates to significant fatigue, and also orthostatic intolerance, and orthostatic hypotension [7–10]."

This sentence:

"...three months after COVID-19 start with symptoms that continue for at least two months and cannot be explained by another diagnosis."

Should be changed to

"three months after COVID-19 infection with symptoms that continue for at least two months and cannot be explained by another diagnosis."

Methods

This sentence:

"In the temporal domain, measures of SDNN, rMSSD, and the proportion of successive RR periods exceeding 50 ms (pNN50)"

Should be changed to:

"In the temporal domain, measures of SDNN, rMSSD, and the proportion of successive RR periods exceeding 50 ms (pNN50) were studied"

Author Response

Dear reviewer

We would like to thank you for your review, which made it possible to considerably improve our article.

Reviewer 4 Report

I am satisfied with the corrections made, and I believe that the paper is of interest to the readers.

Author Response

(The authors gave the same response as above.)
